# Provable Guarantees for Sparsity Recovery with Deterministic Missing Data Patterns

**Chuyang Ke**                                                    *cke@purdue.edu*
*Computer Science*
*Purdue University*

**Jean Honorio**                                                  *jhonorio@unimelb.edu.au*
*School of Computing and Information Systems*
*The University of Melbourne*

**Reviewed on OpenReview:** *https://openreview.net/forum?id=SSqOqAwpN7*

## Abstract

We study the problem of consistently recovering the sparsity pattern of a regression parameter vector from correlated observations governed by deterministic missing data patterns using Lasso. We consider the case in which the observed dataset is censored by a deterministic, non-uniform filter. Recovering the sparsity pattern in datasets with deterministic missing structure can be arguably more challenging than recovering in a uniformly-at-random scenario. In this paper, we propose an efficient algorithm for missing value imputation by utilizing the topological property of the censorship filter. We then provide novel theoretical results for exact recovery of the sparsity pattern using the proposed imputation strategy. Our analysis shows that, under certain statistical and topological conditions, the hidden sparsity pattern can be recovered consistently with high probability in polynomial time and logarithmic sample complexity.

## 1 Introduction

Missing entries in real-world datasets often exhibit deterministic patterns. In federated learning frameworks, sensitive features collected from clients may be censored before being sent to the central server. In electronic health record (EHRs) data, certain lab results may no longer be collected during the postoperative window. Government bureaus may censor certain fields before releasing census data. To deal with missing entries, arguably the most commonly used technique is *data imputation*. Imputation is the process of replacing missing data in a dataset with certain computed values. Common imputation strategies include filling missing entries with row / column mean, median, mode, extreme values, among others. However, most imputation methods do not come with theoretical guarantees. When talking about the quality of imputation methods, prior research mainly evaluates the accuracy boost, before and after imputation, on specific downstream test sets (Wang et al., 2019; Liu & Gopalakrishnan, 2017; Myrtveit et al., 2001). These metrics being used are application-oriented. On the other hand, if we consider imputation itself as the ultimate task alone (i.e., an unsupervised learning task), it is well-known that under low-rank assumptions, matrix completion is possible with theoretical guarantees. The drawbacks are: 1) real world datasets might not follow the missing-at-random assumption in (Candès & Recht, 2009; Candes & Plan, 2010; Recht, 2011) or the structural constraints of the deterministic missingness pattern in (Bhojanapalli & Jain, 2014; Burnwal & Vidyasagar, 2020; Chatterjee, 2020; Lee & Shraibman, 2013; Shapiro et al., 2019; Tsakiris, 2023); and most importantly, 2) matrix completion does not give any guarantee about the downstream tasks utilizing the imputed matrix.

In this paper, we propose the class of *censored supervised learning* tasks, in which the dataset is masked by some deterministic and non-uniform censorship filters. A censorship filter removes certain entries from the true dataset, so that the observed part of the dataset contains missing entries in a deterministic fashion.

Furthermore, we pick *sparsity recovery* as the downstream task in our analysis. Also known as feature selection, sparsity recovery is the task of recovering the *support set* or *sparsity pattern* of a vector $w^* \in \mathbb{R}^p$, from noisy and correlated observations.

It is worth highlighting, that our goal is not to reinvent sparsity recovery or Lasso. The task itself has been extensively studied in the past two decades (Marques et al., 2018; Wainwright, 2009a;b). We also need to highlight, that we are not proposing another heuristic imputation method. Such techniques (filling mean, low rank completion, to name a few) have been proposed and extensively applied in the industry. Instead, we are proposing a unified framework for analyzing the relationship between the sparsity structure in a censored dataset with deterministic missing patterns, and the quality of sparsity recovery, in a formal way with *with provable guarantees*. A censorship filter applied to the dataset brings new challenges from both the algorithmic side (missing data imputation) and the statistical side (sparsity recovery guarantee), and we are interested in the synergy between these two parts.

Here we briefly discuss the implications and the related works.

**Missing Data Techniques.** When dealing with missing values in a dataset, researchers have been using heuristic imputation methods since the first day of machine learning. Such methods include filling missing entries with row / column mean, median, mode, extreme values, among others. Another examples include using random forests for imputation (Stekhoven & Bühlmann, 2012; Van Buuren et al., 1999) and multiple imputation (Carpenter & Kenward, 2012; Murray, 2018). However, it is known that these imputation methods rarely have theoretical guarantees in specific machine learning tasks, including sparsity recovery. Regarding missing data patterns, Fletcher Mercaldo & Blume (2020) proposed the idea of pattern submodels, that is, training a set of submodels for every possible missing value pattern in the observed data. Such approach will be computationally expensive if the missing data pattern is nontrivial. Our goal is to design an imputation method, that is computationally efficient without training multiple submodels, and has theoretical guarantees in the context of sparsity recovery.

**Sparsity Recovery.** The problem of sparsity recovery has been studied extensively during the past 20 years. One of the most widely used algorithm is $l_1$ regularized quadratic programming, also referred to as Lasso. However, most prior literature focus on the fully observed case. For instance, Wainwright (2009a); Meinshausen & Yu (2009) provided theoretical guarantees of sparsity recovery through Lasso when the observation matrix is fully observed. In comparison, the number of works that analyze Lasso given that the dataset is partially observed, is limited. Loh & Wainwright (2015) considered a so-called corruption mechanism, such that every entry in the original dataset is observed with probability $1 - \theta$, and unobserved with probability $\theta$. Nguyen & Tran (2012) proposed a tangentially related model, in which part of the outcome vector is unobserved. The analysis of sparsity recovery guarantee in these cases are usually straightforward, since the pattern of missing entries is uniformly distributed, thus can be viewed as extra noises in the model.

**Randomness in Missing Structure.** It should be highlighted, that the notion of censorship filters in our paper is different from existing discussion of missing data mechanisms in prior literature. This includes definitions such as Missing At Random (MAR), Missing Completely At Random (MCAR), and Missing Not At Random (MNAR) (Mohan et al., 2013; Little & Rubin, 2019). These mechanisms describe how the probability of observing missing entries relate to the values of the underlying true data, whereas our censorship filter is deterministic, arguably more relevant in the real world.

We try to answer the following questions in this paper:

- Does there exists an imputation method for missing entries, so that sparsity recovery algorithms can be applied to the imputed dataset?

- Under what statistical and topological conditions can our workflow correctly and efficiently recover the sparsity pattern?

We propose a simple yet novel sparsity recovery workflow, which 1) imputes the missing entries using their most significant observed neighboring feature, and 2) runs Lasso to recover the sparse pattern using the imputed data. More importantly, our framework can be analyzed rigorously. We provide theoretical

guarantees for the quality of sparsity recovery, in terms of the topological structure of the censorship filter, using the proposed workflow. Our analysis focuses on the case with most significant neighboring feature only, and this can be easily generalized to more neighboring features.

**Summary of Our Contribution.** Our work is mostly theoretical. We provide a series of novel results in this paper:

- We propose a simple yet novel imputation method to fill the missing entries that are censored by an deterministic censorship filter. Our strategy computes the missing value from its most significant neighboring feature, and can be easily generalized to the case of multiple neighboring features.

- We provide provable theoretical guarantees for recovery of the underlying sparsity structure using our imputation method. We analyze the statistical and topological conditions that govern efficient exact recovery. We establish the sample complexity guarantees for our workflow to succeed with high probability. Our theorems also provide guidelines for setting regularization parameters.

## 2 Preliminaries

In this section, we provide the formal setup of our problem and introduce all notations that will be used throughout the paper.

We first introduce the definition of censorship filters. We use $(X, y, M)$ to denote the dataset in a supervised learning task, where $X \in \mathbb{R}^{n \times p}$ is the feature matrix, and $y \in \mathbb{R}^n$ is the label vector. A censorship filter $M \in \{0, 1\}^{n \times p}$ is a binary matrix applied to the feature matrix $X$. For every sample $k$ and feature $i$, $X_{k,i}$ is observed by the learner if and only if $M_{k,i} = 1$. In other words, entries with $M_{k,i} = 0$ are missing and need to be imputed. It is worth highlighting that the censorship filter $M$ is deterministic and non-uniform, i.e., there is no randomness in $M$.

### 2.1 Censored Sparsity Recovery Model

We now present the application of censorship filters to the task of sparsity recovery. Suppose that there exists an unknown fixed vector $w^* \in \mathbb{R}^p$, and $w^*$ is sparse. We denote its support set as $S = \{i \in [p] \mid w_i^* \neq 0\}$, and the cardinality of the support set as $s = |S| \ll p$. Let $X \in \mathbb{R}^{n \times p}$ be the input data generated by nature, such that for every $k \in [n]$, sample $X_{k,:} \in \mathbb{R}^p$ fulfills: 1) zero-mean; 2) with covariance $\Sigma$; 3) each $X_{k,i}$ is sub-Gaussian with parameter $\sigma_X^2 \Sigma_{i,i}$. Then the labels $y \in \mathbb{R}^n$ are generated in the form of

$$y = Xw^* + \epsilon,$$

where $\epsilon \in \mathbb{R}^n$ is the additional zero-mean sub-Gaussian noise with parameter $\sigma_\epsilon^2$. It is known that in the fully observed case, the sparsity recovery of $S$ given $X$ and $y$ can be achieved through solving the following $l_1$ constrained quadratic program, known as Lasso:

$$\underset{w}{\text{minimize}} \qquad \frac{1}{2n}\|Xw - y\|^2 + \lambda\|w\|_1,$$

where $\lambda$ is the regularization parameter. Now a censorship filter $M \in \{0, 1\}^{n \times p}$ is imposed on the learner, such that all entries with $M_{k,i} = 0$ is masked and missing from $X$.

Our task consists of two parts. First, we want to impute $\hat{X}$ from the observed part of $X$, such that $\hat{X}_{k,i} = X_{k,i}$ if $M_{k,i} = 1$. This ensures that the observed entries are not changed. Second, we solve Lasso using the imputed matrix in the form of

$$\underset{w}{\text{minimize}} \qquad \frac{1}{2n}\left\|\hat{X}w - y\right\|^2 + \lambda\|w\|_1, \tag{1}$$

and we claim that the support set recovered by (1) is consistent with the ground truth. We also include the necessary definitions for completeness.

**Definition 1.** *A zero-mean random variable $x$ is sub-Gaussian with parameter $\sigma^2$, if for all $t > 0$, we have $\mathbb{E}\left[\exp(tx)\right] \leq \exp(\sigma^2 t^2/2)$.*

**Definition 2.** *A zero-mean random vector $x = (x_1, \ldots, x_p)$ is sub-Gaussian with parameter $\sigma^2$, if for all $u \in \mathbb{R}^p$ with $\|u\| = 1$, we have $\mathbb{E}\left[\exp(u^\top x)\right] \le \exp(\sigma^2/2)$.*

## 2.2 Notations

Without specification we use lowercase letters (e.g., $a$, $b$, $u$, $v$) for scalars and vectors, and uppercase letters (e.g., $A$, $B$, $C$) for matrices and sets. For any natural number $n$, we use $[n]$ to denote the set $\{1, \ldots, n\}$. We use $\mathbb{R}$ to denote the set of real numbers. We use $\mathbf{1}$ to denote the all-one vector, and $\mathbf{0}$ for the all-zero vector. For any vector $u$, we use $\operatorname{diag}(u)$ to denote the diagonal matrix with $u$ in the diagonal, $\|u\|$ to denote the Euclidean norm, $\|u\|_1$ to denote the $l_1$ norm, and $\|u\|_\infty$ to denote the infinity norm. For any matrix $A$, we use $\lambda_{\min}(A)$ to denote its smallest eigenvalue, $\operatorname{tr}(A)$ to denote its trace, $\|\|A\|\|$ to denote its spectral norm, and $\|\|A\|\|_\infty = \max_i \sum_j |A_{i,j}|$ to denote its $l_\infty$ operator norm. We use $\circ$ to denote the Hadamard product. We use $\operatorname{sgn}(\cdot)$ to denote the sign function.

When dealing with entries in a matrix, we use notation : to denote the whole row or column. For example, $A_{1,:}$ refers to the first row of matrix $A$. We also use index sets in subscripts to select submatrices. For example, $A_{S,S}$ is the submatrix obtained by deleting all rows and columns with indices that are not in the index set $S$ from $A$. When the context is clear, we use single subscripts to denote the choice of columns. An example is that $A_2$ denotes the second column of $A$.

We use $S^{\mathsf{c}}$ to denote the complement of the support set. Similarly, we use $M^{\mathsf{c}}$ to denote the complement of the censorship filter, algebraically $M^{\mathsf{c}} = \mathbf{1}\mathbf{1}^\top - M$.

In our analysis, we use $\Sigma_{\mathrm{dmax}} := \max_i \Sigma_{i,i}$ to denote the maximum diagonal entry in $\Sigma$, and $\Sigma_{\mathrm{dmin}} := \min_i \Sigma_{i,i}$ to denote the minimum.

For distributions, we use subG to denote sub-Gaussian distribution, and subE to denote sub-Exponential distribution.

## 3 Algorithm

In this section, we setup our censored sparsity recovery problem and provide theoretical guarantees. We first introduce the necessary statistical assumptions and definitions.

**Assumption 1** (Positive Definiteness)**.** *We assume that the population covariance matrix $\Sigma$ is positive definite on the support $S$. In particular, we use $\beta := \lambda_{min}(\Sigma_{S,S}) > 0$ to denote its smallest eigenvalue.*

**Assumption 2** (Mutual Incoherence)**.** *We assume that the population covariance matrix $\Sigma$ fulfills the mutual incoherence condition $\left\|\left\|\Sigma_{S^{\mathsf{c}},S} \Sigma_{S,S}^{-1}\right\|\right\|_\infty \le 1 - \gamma$, for some $\gamma \in (0, 1]$.*

Recall that $X \in \mathbb{R}^{n \times p}$ is the feature matrix generated by nature, and $M \in \{0,1\}^{n \times p}$ is the deterministic censorship filter. We use $X_M$ to denote the observed feature matrix, where $(X_M)_{k,i} = X_{k,i}$ if $M_{k,i} = 1$, and $(X_M)_{k,i} = \star$ denotes the missing value otherwise.

Let $H \in \mathbb{R}^{p \times p}$ denote the sample covariance matrix. Since only $X_M$ is observed, zero-mean, and contains missing values, $H_{i,j}$ is computed as $H_{i,j} = \frac{1}{|\{k | M_{i,k} = M_{j,k} = 1\}|} \sum_{k, M_{i,k} = M_{j,k} = 1} X_{k,i} X_{k,j}$.

We use $\zeta_{i,j} := \Sigma_{i,j}^2 / \Sigma_{j,j}$ to denote the (population) neighbor score of two features $i$ and $j$. Intuitively, the neighbor score measures how related two features are. A higher neighbor score indicates that $i$ is more related to $j$. Similarly, we use $\hat{\zeta}_{i,j} := H_{i,j}^2 / H_{j,j}$ to denote the empirical neighbor score. Let $\Pi(i) := \arg\max_{j \in [p] - \{i\}} \hat{\zeta}_{i,j}$ be the top neighbor feature of $i$. The intuition is that if a sample has feature $i$ missing, we use its top neighbor feature $\Pi(i)$ to impute it. To simplify analysis we introduce the following assumption.

**Assumption 3.** *We assume that the top neighbor feature of any missing entry is always observed, that is, we assume $M_{k,\Pi(i)} = 0$ if $M_{k,i} = 1$.*

In practice, one can use the second top feature instead (or third, fourth, etc.), if the top feature is not observed. The assumption only serves to simplify the proofs by reducing the number of concentrations required.

We use $\tau_i$ to denote the (population) error ratio of feature $i$, such that $\tau_i = \frac{\Sigma_{i,\Pi(i)}}{\Sigma_{\Pi(i),\Pi(i)}}$. The motivation is that the error ratio measures how much variance will be gained, if we use the imputed value instead of the true value in the algorithm. Similarly we have the empirical error ratio $\hat{\tau}_i = \frac{H_{i,\Pi(i)}}{H_{\Pi(i),\Pi(i)}}$. We now introduce our censored sparsity recovery algorithm, given the observation of the censored dataset.

---

**Algorithm 1** Censored Sparsity Recovery

---

**Input:** Observed dataset $(X_M, y)$, regularization parameter $\lambda$
**Output:** Imputed feature matrix $\hat{X}$, recovered model vector $\tilde{w}$

1: Compute sample covariance matrix $H$ from $X_M$
2: **for** every feature pair $(i,j) \in [p] \times [p]$ **do**
3:     Compute empirical neighbor score $\hat{\zeta}_{i,j} = H_{i,j}^2/H_{j,j}$
4: **end for**
5: **for** every feature $i \in [p]$ **do**
6:     Compute top neighboring feature $\Pi(i) = \arg\max_{j\in[p]-\{i\}} \hat{\zeta}_{i,j}$
7:     Compute empirical error ratio $\hat{\tau}_i = H_{i,\Pi(i)}/H_{\Pi(i),\Pi(i)}$
8: **end for**
9: Initialize imputation matrix $\bar{X} \in \mathbb{R}^{n \times p}$
10: **for** every entry $(k,i) \in [n] \times [p]$ **do**
11:     $\bar{X}_{k,i} \leftarrow X_{k,\Pi(i)}\hat{\tau}_i$
12: **end for**
13: Compute imputed matrix $\hat{X} = M \circ X + M^c \circ \bar{X}$
14: Solve the following Lasso program

$$\hat{w} = \underset{w}{\text{minimize}} \quad l(w) + \lambda\|w\|_1 \tag{2}$$

$$\text{where} \quad l(w) = \frac{1}{2n}\left\|\hat{X}w - y\right\|^2.$$

---

Algorithm 1 takes the observed dataset as the input and imputes the missing entries given by $\hat{X}$. Understandably the imputed data $\hat{X}$ and the true data $X$ are equivalent on the support of $M$. We use $\Delta$ to denote the imputation error matrix, defined as $\Delta := \hat{X} - X$. Naturally $\Delta_{ij} = 0$ if $M_{ij} = 1$. After that, our algorithm solves the Lasso program (2), and the support of $\hat{w}$ gives the recovered support set $\hat{S}$.

## 4 Guarantees of Censored Sparsity Recovery

### 4.1 Consistency of Imputation through Empirical Score

In this section, we prove consistency of our imputation step, by choosing the top neighboring feature using the empirical neighbor score as in Algorithm 1. The proofs of Theorems and Lemmas can be found in Appendix.

Here is the motivation: in Algorithm 1, we choose $\Pi(i) = \arg\max_{j\in[p]-\{i\}} \hat{\zeta}_{i,j}$ based on the observed samples, where $\hat{\zeta}$ is the empirical score. However, there is no guarantee that the order of $\hat{\zeta}$ is consistent with the underlying true $\zeta$. Our goal is to identify the sufficient conditions, such that $\arg\max_{j\in[p]-\{i\}} \hat{\zeta}_{i,j} = \arg\max_{j\in[p]-\{i\}} \zeta_{i,j}$. Equivalently, it is desirable to ensure that, $\zeta_{i,\Pi(i)} > \zeta_{i,j}$ holds if and only if $\hat{\zeta}_{i,\Pi(i)} > \hat{\zeta}_{i,j}$ holds with high probability for all $j \neq i$.

Our proof relies on the following lemma. The proofs can be found in Appendix.

**Lemma 1.** *For every feature $i \in [p]$, its ratio between the sample variance and population variance fulfills*

$$\mathbb{P}\left\{\frac{1}{2} \leq \frac{H_{i,i}}{\Sigma_{i,i}} \leq \frac{3}{2}\right\} \geq 1 - 4\exp\left(-\frac{n\Sigma_{dmin}^2}{512(1+4\sigma_X^2)^2\Sigma_{dmax}^2}\right).$$

We now present the consistency guarantee for our imputation method proposed in Algorithm 1.

**Theorem 1.** *For every feature $i \in [p]$, if the population neighbor score fulfills*

$$\zeta_{i,\Pi(i)} - 3\zeta_{i,j} > \left|\Sigma_{\Pi(i),\Pi(i)}\right| + 3\left|\Sigma_{j,j}\right| + \Sigma_{dmin}$$

*for every feature $j \neq \Pi(i)$, then the sample neighbor score fulfills*

$$\mathbb{P}\left\{\hat{\zeta}_{i,\Pi(i)} > \hat{\zeta}_{i,j}\right\} \geq 1 - 4\exp\left(-\frac{n\Sigma_{dmin}^2}{512(1 + 4\sigma_X^2)^2\Sigma_{dmax}^2}\right).$$

*Consequently, the imputation result from the empirical score is consistent with the imputation result from the population score with high probability.*

*Remark* 1. In the statement above we have a coefficient of 3. This coefficient is determined by the setting $t = \frac{1}{2}\Sigma_{\text{dmin}}$ in Lemma 1, and can be changed to any constant that is greater but arbitrarily close to 1. This will only affect the constant terms in the high probability statement, and the $1 - O(\exp(-n))$ rate holds.

## 4.2 Primal-dual Witness

We prove the correctness of Algorithm 1 for recovering the true support set $S$, through the primal-dual witness framework and Karush-Kuhn-Tucker (KKT) conditions at the optimum.

**Step 1**: Let $\tilde{w}$ be the solution (primal variable) to the following restricted problem

$$\underset{w_S \in \mathbb{R}^s}{\text{minimize}} \quad l((w_S, \mathbf{0})) + \lambda\|w_S\|_1, \tag{3}$$

with $\tilde{w}_{S^c} = \mathbf{0}$.

**Step 2**: Let $z \in \mathbb{R}^p$ be the dual variable fulfilling the complementary slackness condition on $S$. That is, for every $i \in S$, $z_i = \text{sgn}\left(\tilde{w}_i\right)$ if $\tilde{w}_i \neq 0$, and $z_i \in [-1, +1]$ otherwise.

**Step 3**: Solve for $z_{S^c} \in \mathbb{R}^{p-s}$ to fulfill the following stationarity conditions:

$$[\nabla l((\tilde{w}_S, \mathbf{0}))]_S + \lambda z_S = \mathbf{0} \tag{4}$$

$$[\nabla l((\tilde{w}_S, \mathbf{0}))]_{S^c} + \lambda z_{S^c} = \mathbf{0} \tag{5}$$

**Step 4**: Verify that the strict dual feasibility condition is fulfilled:

$$\|z_{S^c}\|_\infty < 1. \tag{6}$$

Since Step 1 through 3 are constructive, it is sufficient to prove Step 4. If the conditions above are fulfilled, our Algorithm 1 recovers the true support with high probability, i.e., $\hat{S} = S$, where $\hat{S}$ is the support of the recovered vector $\hat{w}$ in (2) and $S$ is the true support.

## 4.3 Verifying Strict Dual Feasibility

To verify the strict dual feasibility condition (6), we first consider the quadratic loss function $l(w)$. Note that

$$
\begin{aligned}
l(w) &= \frac{1}{2n}\left\|\hat{X}w - y\right\|^2 \\
&= \frac{1}{2n}\left\|\hat{X}w - Xw^* - \epsilon\right\|^2 \\
&= \frac{1}{2n}\|(M \circ X + M^c \circ \bar{X})w - (M \circ X + M^c \circ X)w^* - \epsilon\|^2 \\
&= \frac{1}{2n}\|(M \circ X)(w - w^*) + (M^c \circ \bar{X})w - (M^c \circ X)w^* - \epsilon\|^2.
\end{aligned}
$$

We have the gradient

$$\nabla l(w) = \frac{1}{n}\hat{X}^\top((M \circ X)(w - w^*) + (M^c \circ \bar{X})w - (M^c \circ X)w^* - \epsilon),$$

and Hessian $\nabla^2 l(w) = \frac{1}{n} \hat{X}^\top \hat{X}$. In particular, we can define the sample covariance matrix of the imputed matrix as $\hat{H} := \nabla^2 l(w) = \frac{1}{n} \hat{X}^\top \hat{X}$.

We now consider the restricted problem and the stationarity conditions. Expanding (5) leads to

$$\frac{1}{n} \hat{X}_S^\top [(M \circ X)_S (\tilde{w}_S - w_S^*) + (M^{\mathsf{c}} \circ \bar{X})_S \tilde{w}_S - (M^{\mathsf{c}} \circ X)_S w_S^* - \epsilon] + \lambda z_S = \mathbf{0} \,,$$

$$\frac{1}{n} \hat{X}_{S^{\mathsf{c}}}^\top [(M \circ X)_S (\tilde{w}_S - w_S^*) + (M^{\mathsf{c}} \circ \bar{X})_S \tilde{w}_S - (M^{\mathsf{c}} \circ X)_S w_S^* - \epsilon] + \lambda z_{S^{\mathsf{c}}} = \mathbf{0} \,.$$

Next we solve for $z$. On the support set we have

$$
\begin{aligned}
z_S &= -\frac{1}{\lambda n} \hat{X}_S^\top [(M \circ X)_S (\tilde{w}_S - w_S^*) + (M^{\mathsf{c}} \circ \bar{X})_S \tilde{w}_S - (M^{\mathsf{c}} \circ X)_S w_S^* - \epsilon] \\
&= -\frac{1}{\lambda n} \hat{X}_S^\top [(M \circ X)_S (\tilde{w}_S - w_S^*) + (M^{\mathsf{c}} \circ \bar{X})_S (\tilde{w}_S - w_S^*) + (M^{\mathsf{c}} \circ \bar{X} - M^{\mathsf{c}} \circ X)_S w_S^* - \epsilon] \\
&= -\frac{1}{\lambda n} \hat{X}_S^\top [\hat{X}_S (\tilde{w}_S - w_S^*) + (M^{\mathsf{c}} \circ \bar{X} - M^{\mathsf{c}} \circ X)_S w_S^* - \epsilon] \\
&= -\frac{1}{\lambda n} \hat{X}_S^\top [\hat{X}_S (\tilde{w}_S - w_S^*) + \Delta_S w_S^* - \epsilon] \,.
\end{aligned}
$$
(7)

Similarly on the complement set we have

$$z_{S^{\mathsf{c}}} = -\frac{1}{\lambda n} \hat{X}_{S^{\mathsf{c}}}^\top \left[ \hat{X}_S (\tilde{w}_S - w_S^*) + \Delta_S w_S^* - \epsilon \right] \,. \tag{8}$$

Rearranging the terms in (7) leads to $\tilde{w}_S - w_S^* = -(\hat{X}_S^\top \hat{X}_S)^{-1} \left( \hat{X}_S^\top (\Delta_S w_S^* - \epsilon) + \lambda n z_S \right)$. Plugging the last equation into (8), we obtain $z_{S^{\mathsf{c}}} = z^{(a)} + z^{(b)}$, where we use the shorthand notation

$$z^{(a)} := \frac{1}{\lambda n} \hat{X}_{S^{\mathsf{c}}}^\top \left( I - \hat{X}_S (\hat{X}_S^\top \hat{X}_S)^{-1} \hat{X}_S^\top \right) (\epsilon - \Delta_S w_S^*) \,, \tag{9}$$

$$z^{(b)} := \hat{X}_{S^{\mathsf{c}}}^\top \hat{X}_S (\hat{X}_S^\top \hat{X}_S)^{-1} z_S \,. \tag{10}$$

It remains to verify the strict dual feasibility condition $\|z_{S^{\mathsf{c}}}\|_\infty = \|z^{(a)} + z^{(b)}\|_\infty < 1$. This can be further broken down into two parts: we first prove that $\|z^{(a)}\|_\infty < \gamma/4$, and then prove $\|z^{(b)}\|_\infty \le 1 - \gamma/4$.

### 4.4 Bound of $z^{(a)}$

In this section we analyze the upper bound of $\|z^{(a)}\|_\infty$. For every feature $i \in S^{\mathsf{c}}$ and sample $k \in [n]$, we define the following variance proxy

$$h_k^2 := \sigma_\epsilon^2 + \sigma_X^2 \sum_{i \in S} (\hat{\tau}_i^2 \Sigma_{\Pi(i),\Pi(i)} + \Sigma_{i,i}) M_{k,i}^{\mathsf{c}} (w_i^*)^2 \,,$$

and

$$g_k(i)^2 := M_{k,i} \sigma_X^2 \Sigma_{i,i} + (1 - M_{k,i}) \frac{9}{4} \sigma_X^2 \Sigma_{\Pi(i),\Pi(i)} \tau_i^2 \,,$$

assuming $h_k, g_k(i) \ge 0$. We also denote the maximum variance proxy as

$$h_{\max} = \max_{k \in n} h_k \,, \qquad g_{\max} = \max_{k \in n} \max_{i \in S^{\mathsf{c}}} g_k(i) \,,$$

across all sample $k \in [n]$. We now provide the statement of the theorem.

**Theorem 2.** *By setting the regularization parameter*

$$\lambda > 20 h_{\max} \cdot g_{\max} / \gamma \,,$$

*we have* $\|z^{(a)}\|_\infty < \gamma/4$ *with probability at least* $1 - O((p - s) \exp(-n))$.

*Remark* 2. Our proofs of Theorem 2 and Theorem 3 to bound $z^{(a)}$ and $z^{(b)}$, rely on the careful analysis of the sub-Gaussian condition of the imputed matrix. Techniques from prior literature do not work in our model, because our missing structure $M$ is deterministic and cannot be reduced to some uniformly random noise. For instance, one classic technique is to write $X_{S^c}$ as a predictor of $X_S$ using the conditional covariance matrix (Wainwright, 2009a). This does not work in our case, because $\hat{X}$ (imputed matrix) will not cancel with the complement projection on $X$ (original matrix).

*Remark* 3. One may note that the magnitude of $\|\epsilon - \Delta_S w_S^*\|$ is directly related to the quality of regression in the proof above. Intuitively, the whole term measures the noise level in our algorithm: $\epsilon$ for the noise generated by nature, and $\Delta_S w_S^*$ for the imputation noise, consisting of the imputation error on the support $\Delta_S$ and the ground truth $w_S^*$. This provides the insight, that if the magnitude of $w^*$ is large, censored sparsity recovery will be harder because of a higher imputation noise level.

*Proof.* Here we consider every feature $j \in S^c$. It is worth noting that by definition, $\hat{X}_S(\hat{X}_S^\top \hat{X}_S)^{-1}\hat{X}_S^\top$ is an orthogonal projection matrix to the column space of $\hat{X}$, thus for simplicity, we denote the projection $P_{\hat{X}_S} := \hat{X}_S(\hat{X}_S^\top \hat{X}_S)^{-1}\hat{X}_S^\top$.

Using Cauchy-Schwarz inequality and the fact that the norm of a orthogonal projection matrix is bounded above by 1, we obtain

$$\left|z_j^{(a)}\right| = \left|\sum_{k=1}^n \hat{X}_{k,j}\left[(I - P_{\hat{X}_S})\left(\frac{1}{\lambda n}(\epsilon - \Delta_S w_S^*)\right)\right]_k\right|$$

$$\leq \left\|(I - P_{\hat{X}_S})\left(\frac{1}{\lambda n}(\epsilon - \Delta_S w_S^*)\right)\right\| \cdot \left\|\hat{X}_j\right\|$$

$$\leq \left\|\frac{1}{\lambda n}(\epsilon - \Delta_S w_S^*)\right\| \cdot \left\|\hat{X}_j\right\|$$

$$= \frac{1}{\lambda n}\|\epsilon - \Delta_S w_S^*\| \cdot \left\|\hat{X}_j\right\|.$$

We proceed to bound $\|\epsilon - \Delta_S w_S^*\|$ for each entry. For every sample $k \in N$, we have

$$\epsilon_k - \Delta_{k,S} w_S^* = \epsilon_k - \sum_{i \in S} \Delta_{k,i} w_i^*$$

$$= \epsilon_k - \sum_{i \in S} (\hat{X}_{k,i} - X_{k,i}) w_i^*$$

$$= \epsilon_k - \sum_{i \in S} (\bar{X}_{k,i} - X_{k,i}) M_{k,i}^c w_i^*$$

$$= \epsilon_k - \sum_{i \in S} (\hat{\tau}_i X_{k,\Pi(i)} - X_{k,i}) M_{k,i}^c w_i^*.$$

Under the assumption of $X_{k,i} \sim \text{subG}(\sigma_X^2 \Sigma_{i,i})$, $\epsilon_k \sim \text{subG}(\sigma_\epsilon^2)$, note that the entrywise imputation error $\hat{\tau}_i X_{k,\Pi(i)} - X_{k,i}$ is sub-Gaussian with parameter $(\hat{\tau}_i^2 \Sigma_{\Pi(i),\Pi(i)} + \Sigma_{i,i})\sigma_X^2$. As a result, $\epsilon_k - \Delta_{k,S} w_S^*$ is sub-Gaussian with parameter $h_k^2$.

Since samples are independently generated across all $k$'s, we know that $\epsilon - \Delta_S w_S^*$ is a sub-Gaussian vector with parameter at most $h_{\max}^2$, where $h_{\max} := \max_{k \in [n]} h_k$. Then, by Lemma 4, for all $t > 0$ we have

$$\mathbb{P}\left\{\|\epsilon - \Delta_S w_S^*\|^2 > h_{\max}^2(n + 2\sqrt{nt} + 2t)\right\} \leq e^{-t}.$$

Setting $t = n$ and taking square roots, this leads to

$$\mathbb{P}\left\{\|\epsilon - \Delta_S w_S^*\| > h_{\max}\sqrt{5n}\right\} \leq e^{-n}.$$

Next we bound $\left\|\hat{X}_j\right\|$. For every sample $k$, $\hat{X}_{k,j}$ is sub-Gaussian with parameter $\sigma_X^2 \Sigma_{j,j}$ if $M_{k,j} = 1$, or with parameter $\sigma_X^2 \Sigma_{\Pi(j),\Pi(j)} \hat{\tau}_j^2$ otherwise. In particular, the latter is bounded by $\frac{9}{4}\sigma_X^2 \Sigma_{\Pi(j),\Pi(j)} \tau_j^2$ with probability at least $1 - O(\exp(-n))$ using Lemma 5. Put together, $\hat{X}_{k,j}$ is sub-Gaussian with parameter at most $g_k(j)^2$. Since samples are independently generated across all $k$'s, we know that $\hat{X}_j$ is a sub-Gaussian vector with parameter at most $g_{\max}(j)^2 := \max_{k \in n} g_k(j)^2$. Then, by Lemma 4, for all $t > 0$ we have

$$\mathbb{P}\left\{ \left\|\hat{X}_j\right\|^2 > g_{\max}(j)^2(n + 2\sqrt{nt} + 2t) \right\} \leq \mathrm{e}^{-t}\,.$$

Setting $t = n$, this leads to

$$\mathbb{P}\left\{ \left\|\hat{X}_j\right\| > g_{\max}(j)\sqrt{5n} \right\} \leq \mathrm{e}^{-n}\,.$$

Combining both parts above, with probability at least $1 - O(\exp(-n))$, we require that

$$\begin{aligned}
\left|z_j^{(a)}\right| &\leq \frac{1}{\lambda n}\left\|\epsilon - \Delta_S w_S^*\right\| \cdot \left\|\hat{X}_j\right\| \\
&\leq \frac{1}{\lambda n} h_{\max}\sqrt{5n} \cdot g_{\max}(j)\sqrt{5n} \\
&= \frac{5h_{\max}g_{\max}(j)}{\lambda}\,.
\end{aligned}$$

Our goal is to ensure that $\left|z_j^{(a)}\right|$ is less than $\gamma/4$ for all $j \in S^{\mathsf{c}}$. Thus, the high probability sufficient condition is

$$\lambda > 20h_{\max} \cdot g_{\max}(j)/\gamma\,.$$

Taking a union bound for all $j \in S^{\mathsf{c}}$ leads to the final result. $\qquad\square$

### 4.5  Bound of $z^{(b)}$

Here we provide the upper bound for $z^{(b)}$. Our analysis relies on the following auxillary lemma.

**Lemma 2.** *Under the mild condition $\gamma < 6/7$, the sample covariance matrix $H$ fulfills the mutual incoherence condition*

$$\left\|\left\|H_{S^{\mathsf{c}},S}H_{S,S}^{-1}\right\|\right\|_\infty \leq 1 - \gamma/2\,,$$

*with probability at least $1 - O\left(s(p-s)\exp\left(-\frac{\beta^2\gamma^2 n}{s^3}\right)\right) - O\left(s^2\exp\left(-\frac{\beta^2\gamma^2 n}{s^3(1-\gamma)^2}\right)\right)$.*

**Theorem 3.** *Under the mild condition $\gamma < 6/7$, we have $\left\|z^{(b)}\right\|_\infty \leq 1 - \gamma/4$ with probability at least $1 - O\left(s(p-s)\exp\left(-\frac{\beta^2\gamma^2 n}{s^3}\right)\right) - O\left(s^2\exp\left(-\frac{\beta^2\gamma^2 n}{s^3(1-\gamma)^2}\right)\right) - O\left(s^2\exp\left(-\frac{\beta^2\gamma^2 n}{s^3(1-\gamma/2)^2}\right)\right)$.*

*Proof.* Note that

$$\begin{aligned}
\left\|z^{(b)}\right\|_\infty &= \left\|\hat{X}_{S^{\mathsf{c}}}^\top \hat{X}_S(\hat{X}_S^\top \hat{X}_S)^{-1}z_S\right\|_\infty \\
&\leq \left\|\left\|\hat{X}_{S^{\mathsf{c}}}^\top \hat{X}_S(\hat{X}_S^\top \hat{X}_S)^{-1}\right\|\right\|_\infty \|z_S\|_\infty \\
&\leq \left\|\left\|\hat{X}_{S^{\mathsf{c}}}^\top \hat{X}_S(\hat{X}_S^\top \hat{X}_S)^{-1}\right\|\right\|_\infty \\
&\leq \left\|\left\|H_{S^{\mathsf{c}},S}H_{S,S}^{-1}\right\|\right\|_\infty + \left\|\left\|(\hat{H}_{S^{\mathsf{c}},S} - H_{S^{\mathsf{c}},S})H_{S,S}^{-1}\right\|\right\|_\infty \\
&\quad + \left\|\left\|H_{S^{\mathsf{c}},S}(\hat{H}_{S,S}^{-1} - H_{S,S}^{-1})\right\|\right\|_\infty + \left\|\left\|(\hat{H}_{S^{\mathsf{c}},S} - H_{S^{\mathsf{c}},S})(\hat{H}_{S,S}^{-1} - H_{S,S}^{-1})\right\|\right\|_\infty\,.
\end{aligned}$$

We use the shorthand notation to denote the last four terms above, where $HH_1 := H_{S^{\mathsf{c}},S}H_{S,S}^{-1}$, $HH_2 := (\hat{H}_{S^{\mathsf{c}},S} - H_{S^{\mathsf{c}},S})H_{S,S}^{-1}$, $HH_3 := H_{S^{\mathsf{c}},S}(\hat{H}_{S,S}^{-1} - H_{S,S}^{-1})$, and $HH_4 := (\hat{H}_{S^{\mathsf{c}},S} - H_{S^{\mathsf{c}},S})(\hat{H}_{S,S}^{-1} - H_{S,S}^{-1})$, respectively.

Next we bound $\||HH_1\||_\infty$ through $\||HH4\||_\infty$. Regarding $HH_1$, by Lemma 2, with probability at least $1 - O\left(s(p-s)\exp\left(-\frac{\beta^2\gamma^2 n}{s^3}\right)\right) - O\left(s^2 \exp\left(-\frac{\beta^2\gamma^2 n}{s^3(1-\gamma)^2}\right)\right)$, we have $\||HH_1\||_\infty \le 1 - \gamma/2$.

For $HH_2$, using Lemma 7, with probability at least $1 - O\left(s^2 \exp\left(-\frac{\beta^2\gamma^2 n}{s^3(1-\gamma/2)^2}\right)\right)$, we have

$$
\begin{aligned}
\||HH_2\||_\infty &\le \left\|\left|\hat{H}_{S^c,S} - H_{S^c,S}\right\|\right|_\infty \left\|\left|H_{S,S}^{-1}\right\|\right|_\infty \\
&\le \sqrt{s}\left\|\left|\hat{H}_{S^c,S} - H_{S^c,S}\right\|\right|_\infty \left\|\left|H_{S,S}^{-1}\right\|\right| \\
&\le \frac{2\sqrt{s}}{\beta}\left\|\left|\hat{H}_{S^c,S} - H_{S^c,S}\right\|\right|_\infty \\
&\le \frac{\gamma}{12}.
\end{aligned}
$$

Similarly for $HH_3$, with probability of the same order we have

$$
\begin{aligned}
\||HH_3\||_\infty &\le \left\|\left|H_{S^c,S}(\hat{H}_{S,S}^{-1} - H_{S,S}^{-1})\right\|\right|_\infty \\
&\le \left\|\left|H_{S^c,S}(H_{S,S}^{-1}(H_{S,S} - \hat{H}_{S,S})\hat{H}_{S,S}^{-1})\right\|\right|_\infty \\
&\le \left\|\left|H_{S^c,S}H_{S,S}^{-1}\right\|\right|_\infty \left\|\left|H_{S,S} - \hat{H}_{S,S}\right\|\right|_\infty \left\|\left|\hat{H}_{S,S}^{-1}\right\|\right|_\infty \\
&\le \frac{4\sqrt{s}(1-\gamma/2)}{\beta}\left\|\left|H_{S,S} - \hat{H}_{S,S}\right\|\right|_\infty \\
&\le \frac{\gamma}{12}.
\end{aligned}
$$

For $HH_4$, with probability of the same order we have

$$
\begin{aligned}
\||HH_4\||_\infty &\le \left\|\left|\hat{H}_{S^c,S} - H_{S^c,S}\right\|\right|_\infty \left\|\left|\hat{H}_{S,S}^{-1} - H_{S,S}^{-1}\right\|\right|_\infty \\
&\le \frac{8s}{\beta^2}\left\|\left|\hat{H}_{S^c,S} - H_{S^c,S}\right\|\right|_\infty \left\|\left|H_{S,S} - \hat{H}_{S,S}\right\|\right|_\infty \\
&\le \frac{\gamma}{12},
\end{aligned}
$$

where the last inequality holds if $\gamma^2 \le 12\gamma(1-\gamma/2)$, which is always true since $\gamma$ is bounded between 0 and 1.

Combining all four terms above using a union bound, with probability at least $1 - O\left(s(p-s)\exp\left(-\frac{\beta^2\gamma^2 n}{s^3}\right)\right) - O\left(s^2 \exp\left(-\frac{\beta^2\gamma^2 n}{s^3(1-\gamma)^2}\right)\right) - O\left(s^2 \exp\left(-\frac{\beta^2\gamma^2 n}{s^3(1-\gamma/2)^2}\right)\right)$, we have $\left\|z^{(b)}\right\|_\infty \le 1 - \frac{\gamma}{2} + \frac{\gamma}{12} + \frac{\gamma}{12} + \frac{\gamma}{12} = 1 - \frac{\gamma}{4}$. $\square$

### 4.6 Main Result

Armed with the previous results, we present our main contribution. That is, we show that Algorithm 1 recovers the true support with high probability.

**Theorem 4.** *Under the mild condition $\gamma < 6/7$ and by setting the regularization parameter*

$$
\lambda > 20 h_{\max} \cdot g_{\max}/\gamma,
$$

*Algorithm 1 recovers the true support set with high probability, i.e., $\hat{S} = S$, where $\hat{S}$ is the support of the recovered vector $\hat{w}$ in (2) and $S$ is the true support.*

*Proof.* Straightforwardly, by considering the discussion on Section 4.2 and Section 4.3, as well as by invoking Theorem 1, Theorem 2, Theorem 3 and Lemma 6. $\square$

## 5   Discussions

In this section, we validate the proposed Algorithm 1 through synthetic experiments.

**Experiment 1**: We test four imputation strategies in the task of censored sparsity recovery, including our method, imputation by zero, imputation by mean, and imputation by median. We generate $w^*$ such that features in the support are randomly drawn in $[-1, -0.25] \cup [0.25, 1]$. $X$ is generated from Gaussian distribution, with mean $\mathbf{0}$ and covariance $\Sigma$. We set the diagonal of $\Sigma$ to 1, and the off-diagonal to 0.8. We control the number of samples $n = 1000$, number of features $p = 50$, and size of support $s = 10$. The variable is the percentage of missing entries in observed $\hat{X}$. We plot the probability of censored sparsity recovery $\mathbb{P}\left\{\hat{S} = S\right\}$ and the $l_\infty$ distance $\|\hat{w} - w^*\|_\infty$ between the recovered and the true vector, against the percentage of missing entries, in Figure 1a and Figure 1b, respectively. Each trial is run 100 times. It can be seen that our imputation strategy outperforms the others on both metrics.

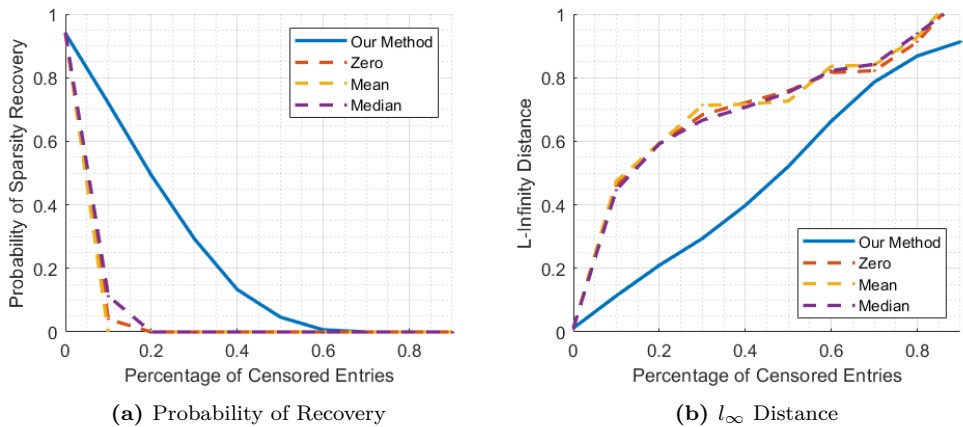

**(a)** Probability of Recovery          **(b)** $l_\infty$ Distance

**Figure 1:** Validation across different numbers of missing entries. Our algorithm achieved recovery when 20% of entries are censored with probability at least one half, while other approaches failed with high probability. The vector recovered by our algorithm is also closer to the ground truth.

**Experiment 2**: We fix the percentage of missing entries to be 20%, and run the same experiment with different $n$, $p$, and $s$. To present the results, we define a weighted constant $C := \log\left(\frac{n}{s^3 \log s(p-s)}\right)$, which is derived from the high probability bound in Theorem 2 and 3. We plot the probability of censored sparsity recovery $\mathbb{P}\left\{\hat{S} = S\right\}$ and the $l_\infty$ distance $\|\hat{w} - w^*\|_\infty$ between the recovered and the true vector, against $C$ in Figure 2a and Figure 2b, respectively. Each trial is run 100 times. From Figure 2a, one can see that our method achieved recovery with probability tending to 1 if $C$ is large enough. This matches our prediction in Theorem 2 and 3.

**Experiment 3**: One of our contributions, is that we focus on the case when the missing data pattern is deterministic. As highlighted above, our analysis provides exact sparsity recovery guarantees when the missing data admit a nontrivial deterministic pattern. In contrast, low-rank matrix completion either assumes that the entries are missing at random, or imposes some structural constraints to the deterministic missingness pattern, both of which might not be followed by real world datasets. To illustrate this, we consider the case where the observed entries follow a chain graph pattern (Figure 3a), where each white entry is observed and black entry is not observed. Note that the features on the two sides are not observed at the same time. This is common in many real world scenarios. For example, in medical data, certain lab tests serve the same purpose and thus are not conducted at the same time.

In this experiment, we demonstrate that our workflow performs better than low-rank matrix completion, when the observed entries follow a deterministic chain graph pattern as in Figure 3a. The chain width ranges from 2 to 20. We control the parameters by setting $n = 200$, $p = 50$, and $s = 10$. The other settings are the same as in previous experiments. We plot the $l_\infty$ distance $\|\hat{w} - w^*\|_\infty$ between the recovered and the true vector,

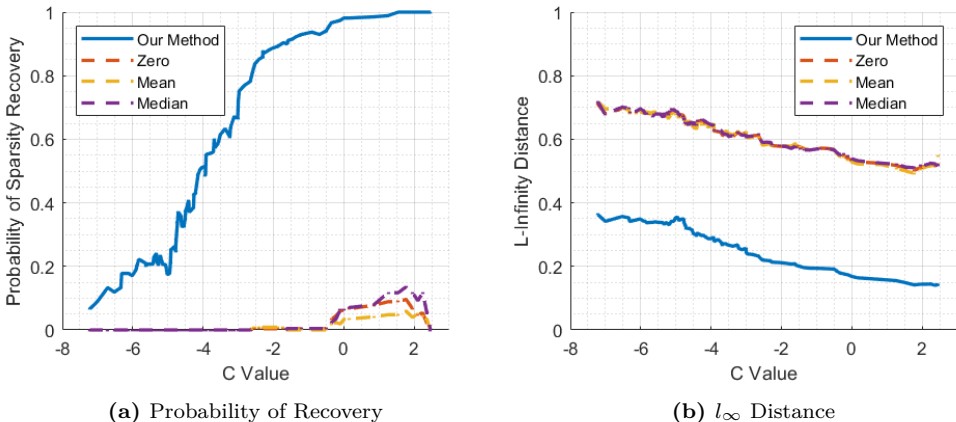

**(a)** Probability of Recovery

**(b)** $l_\infty$ Distance

**Figure 2:** Validation across different $C$, a weighted sample size. Our algorithm achieved recovery $C$ is large enough, matching our theoretical findings.

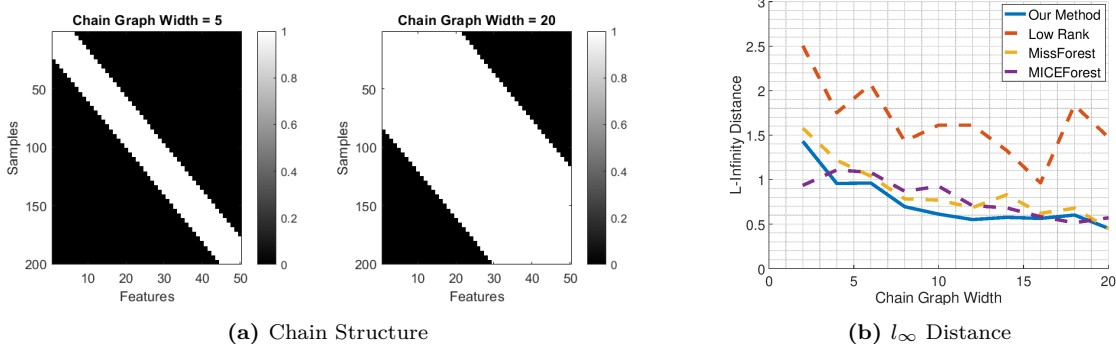

**(a)** Chain Structure

**(b)** $l_\infty$ Distance

**Figure 3:** Validation across different $C$, a weighted sample size. Our algorithm achieved recovery $C$ is large enough, matching our theoretical findings.

against the chain width in Figure 3b. Each trial is run 10 times. From Figure 3b, one can see that our method is more robust than the matrix completion approach when the dataset admits a deterministic chain graph pattern. MissForest (Stekhoven & Bühlmann, 2012) and MICEForest (Van Buuren et al., 1999) are better than matrix completion, but slightly worse than our relatively simpler imputation technique. On the other hand, MissForest and MICEForest are far more complex to study theoretically. Thus, our relatively simpler imputation technique allows for good experimental results, together with a strong theoretical guarantee of support recovery.

# 6 Concluding Remarks

In this paper we proposed the idea of censored supervised learning, in which a censorship filter masks the dataset in a deterministic, non-uniform way. We analyzed the specific case of censored sparsity recovery, and provided imputation strategies and theoretical guarantees.

We currently use the top neighboring feature to impute the missing value in our algorithm. As a future direction, this can be extended to the second, third, ..., neighboring features in a weighted fashion. Another possible option is to take $y$ into account in the imputation step.

Moreover, it would be interesting to see if similar strategies and analysis will follow, for other supervised learning tasks masked by some censorship filters. Most of these problems, though commonly encountered in real world applications, do not have theoretical guarantees about imputation quality.

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
