# OpenReview forum: "Provable Guarantees for Sparsity Recovery with Deterministic Missing Data Patterns"
_TMLR — Accepted by TMLR_

### Review · Reviewer_7JCj · 2023-09-21

**Summary Of Contributions:**

This paper considers the problem of sparse linear regression with missing data. Specifically, the objective is to estimate the support of the sparse regression vector $w \in \mathbb{R}^p$ from knowledge of the measurements $y \in \mathbb{R}^n$ together with an incomplete version $X \odot M$ of the input data matrix $X \in \mathbb{R}^{n \times p}$; here $M$ is a deterministic $\{0,1\}$-matrix that can be thought of as a masking filter. The paper develops theoretical guarantees for the successful recovery of the support of $w$, upon running Lasso on $(\hat{X},y)$, where $\hat{X}$ is an imputed version of the incomplete matrix $X \odot M$, obtained by a simple process.

**Audience:**

Yes

**Broader Impact Concerns:**

-

**Claims And Evidence:**

Yes

**Requested Changes:**

1. Please revise the writing regarding A) and B) above.

2. The output of Algorithm 1 is the imputed matrix \hat{X} and the estimated model vector \tilde{w}. Thus when it is written in the beginning of section 4.2 that the correctness of Algorithm 1 is going to be proved, the reader understandably may think that error bounds for the imputation error and the estimation of w are going to be given. However, all that is being done is to prove that the support of \tilde{w} is the correct one. Please revise the writing to explain what is you mean by "We prove the correctness of Algorithm 1".

3. This is a comment that was given in prior reviews but ignored by the authors: There should be a theorem saying that "if conditions such and such hold, then the support of \tilde{w} given by Algorithm 1 is equal to the support of \tilde{w*}". Currently no such theorem exists.

**Strengths And Weaknesses:**

This is a paper whose prior versions I have reviewed twice, once for NeurIPS '22 and once for ICML '23. I personally find the problem interesting and important, and the theoretical contribution worth of publication. Also, the presentation of the current manuscript has been improved with respect to earlier versions (this was one of the main critiques).

The authors seem to have ignored several suggestions that were given to them, such as 1) specific suggestions for improving clarity, 2) discussing the literature more deeply, and 3) improving the strength of their experiments. Since the contribution is mostly theoretical and the TMLR insists on clarity, 1) and 2) are more serious than 3). Regarding clarity 1) please see the easily fixed points 2 and 3 in the next box. Regarding literature 2), I am very surprised that the authors have chosen to persist on the following writing (even though this was pointed out to them by multiple reviewers), which gives a false impression to the non-expert reader but also does injustice to several existing works on low-rank matrix completion which do give theoretical guarantees under deterministic observation patterns:

A) First paragraph of the introduction (page 1): "...matrix completion is possible with theoretical guarantees (Candes & Recht, 2009). The drawbacks are: 1) the missing-uniformly-at-random assumption is highly ideal for real-world datasets and ..."
B) Experiment 3 (page 12): "In contrast, low rank matrix completion assumes that the entries are missing uniformly at random, which is highly ideal for many real world datasets."

---

> ### Author Response · Authors · 2023-11-15
>
> > I personally find the problem interesting and important, and the theoretical contribution worth of publication.
>
> We thank the reviewer for the appreciation of our work.
>
> > I am very surprised that the authors have chosen to persist on the following writing (even though this was pointed out to them by multiple reviewers), which gives a false impression to the non-expert reader but also does injustice to several existing works on low-rank matrix completion which do give theoretical guarantees under deterministic observation patterns:
> >
> > A) First paragraph of the introduction (page 1): "...matrix completion is possible with theoretical guarantees (Candes & Recht, 2009). The drawbacks are: 1) the missing-uniformly-at-random assumption is highly ideal for real-world datasets and ..." B) Experiment 3 (page 12): "In contrast, low rank matrix completion assumes that the entries are missing uniformly at random, which is highly ideal for many real world datasets."
> >
> > Change 1. Please revise the writing regarding A) and B) above.
>
> Thanks for pointing this out. We will revise the paper in this respect. The following are some matrix completion papers considering non-random missingness in their theoretical analysis, but still using nuclear-norm minimization as an algorithm, which we experimentally use for comparison:
> - Bhojanapalli, S. and Jain, P. (2014). Universal matrix completion. In International Conference on Machine Learning, pages 1881-1889. PMLR.
> - Burnwal, S. P. and Vidyasagar, M. (2020). Deterministic completion of rectangular matrices using asymmetric ramanujan graphs: Exact and stable recovery. IEEE Transactions on Signal Processing, 68:3834-3848.
> - Chatterjee, S. (2020). A deterministic theory of low rank matrix completion. IEEE Transactions on Information Theory, 66(12):8046-8055.
> - Lee, T. and Shraibman, A. (2013). Matrix completion from any given set of observations. Advances in Neural Information Processing Systems, 26.
> - Shapiro, A., Xie, Y., and Zhang, R. (2018). Matrix completion with deterministic pattern: A geometric perspective. IEEE Transactions on Signal Processing, 67(4):1088-1103.
>
> > Change 2. The output of Algorithm 1 is the imputed matrix \hat{X} and the estimated model vector \tilde{w}. Thus when it is written in the beginning of section 4.2 that the correctness of Algorithm 1 is going to be proved, the reader understandably may think that error bounds for the imputation error and the estimation of w are going to be given.
>
> We will state "We prove the correctness of Algorithm 1 for recovering the true support".
>
> > Change 3. This is a comment that was given in prior reviews but ignored by the authors: There should be a theorem saying that "if conditions such and such hold, then the support of \tilde{w} given by Algorithm 1 is equal to the support of \tilde{w*}". Currently no such theorem exists.
>
> Thanks for pointing this out. Currently, we mention at the end of Section 4.2 the conditions for which $\hat{S} = S$ where $\hat{S}$ is the estimated support and $S$ is the true support. We then focus on proving such conditions in our current lemmas/theorems. We wrote the derivations in Section 4.2 and 4.3 (primal-dual witness) without presenting them as lemma/theorem on purpose, to avoid confusion about novelty. (See, e.g., the comment and response to Reviewer GeJb).
>
> Having said that, we agree with the reviewer. Note that our assumptions have been clearly stated independently from lemmas/theorems. We will include a new theorem with the final result $\hat{S} = S$. The proof of this theorem is straightforward and can be done in a couple of lines, by invoking our current lemmas/theorems.

---

### Review · Reviewer_SnfY · 2023-10-19

**Summary Of Contributions:**

The authors propose an algorithm to input missing data.
Then, under standard restricted strong convexity and mutual incoherence assumption, they derive sparse recovery properties when a Lasso is applied to the inputted data.
The authors validate their claims on synthetic data: they empirically show that their algorithm yields better sparsity recovery compared to the usual mean and median imputation techniques.

**Audience:**

Yes

**Claims And Evidence:**

Yes

**Requested Changes:**

- Perform a literature review on inpainting and compare with previous results
- At least add MICE and Miss Forest in the experiments
- A standard inpainting technique would be nice as well

**Strengths And Weaknesses:**

My biggest concern is the considered setting: deterministic censorship filter instead of "usual" missing data at random.
As stated in the article this setting is very relevant, and that's why it has already been extensively studied under the name of inpainting. It seems to me that the article is completely missing this part of the literature, see for instance [3] or [4] for a review.

Experimentally:
- As far as I understand, the imputation algorithm is completely decorrelated from the algorithm applied after, so why not compare it to very standard imputation techniques such as MissForest [1] or MICE-Forest [2]

- Lasso techniques are usually applied when the number of feature $p$ is much larger than the number of samples $n$, hence the proposed imputation technique seems to require $n \times p^2$ compute

- If all the features are normalized, i.e., $\| X_j \|$, it seems to me that the feature that will correlate the most with $j$, will be $j$, and thus assumption 3 requires the missing entry to be observed. Could you comment on this?


[1] Stekhoven, D.J. and Bühlmann, P., 2012. MissForest—non-parametric missing value imputation for mixed-type data. Bioinformatics, 28(1), pp.112-118.

[2] Van Buuren, S., Boshuizen, H.C. and Knook, D.L., 1999. Multiple imputation of missing blood pressure covariates in survival analysis. Statistics in medicine, 18(6), pp.681-694.

[3] Yun, T., Jung, H. and Son, J., 2023. Imputation as Inpainting: Diffusion models for SpatioTemporal Data Imputation.

[4] Elharrouss, O., Almaadeed, N., Al-Maadeed, S. and Akbari, Y., 2020. Image inpainting: A review. Neural Processing Letters, 51, pp.2007-2028.

---

> ### Author Response · Authors · 2023-11-16
>
> > Weakness: ... it has already been extensively studied under the name of inpainting... the article is completely missing this part of the literature, see for instance [3] or [4] for a review.
> >
> > Change: Perform a literature review on inpainting and compare with previous results
> >
> > Change: ...in the experiments. A standard inpainting technique would be nice as well
>
> Our paper performs a theoretical study of sparse regression with missing data in general datasets. Inpainting techniques [3] and [4] are for spatio-temporal datasets, such as images and time sequences. We are not focusing on spatio-temporal data. In general datasets, features do not have a spatial relationship (as in pixels for instance, where a pixel is a neighbor of another pixel). Moreover, in general datasets, samples can be arbitrarily permuted without affecting the final result of regression, unlike in temporal data analysis, where samples are related to a timepoint.
>
> Finally, we are not aware of the use of inpainting for support recovery in sparse regression.
>
> > Weakness: As far as I understand, the imputation algorithm is completely decorrelated from the algorithm applied after, so why not compare it to very standard imputation techniques such as MissForest [1] or MICE-Forest [2]
> >
> > Change: At least add MICE and Miss Forest in the experiments
>
> Thanks for the opportunity to clarify our rationale. For comparison purposes, we use missing data imputation methods with some sort of theoretical guarantees, i.e., matrix completion methods, and avoided using heuristics.
>
> To run our algorithm, we first run the imputation step, and then we run the sparse regression step on the previously imputed data. Given this, it seems these two steps are somewhat independent, but they are in fact tightly related in the sense that our support recovery guarantee relies on both steps.
>
> Having said that, we have tried MissForest and MICEForest in the experiment in Figure 3(b). MissForest and MICEForest are better than matrix completion, but slightly worse than our relatively simpler imputation technique. On the other hand, MissForest and MICEForest are far more complex to study theoretically. Thus, our relatively simpler imputation technique allows for good experimental results, together with a strong theoretical guarantee of support recovery.
>
> > Weakness: Lasso techniques are usually applied when the number of feature $p$ is much larger than the number of samples $n$, hence the proposed imputation technique seems to require $n \times p^2$ compute
>
> Our imputation technique takes at most $n p (p-1)/2$ time if run on a single computer, but it can be easily parallelizable into at most $p (p-1)/2$ different computations of the sample covariances $H_{i,j}$'s from the observed entries only, and neighbor scores $\zeta_{i,j}$'s.
>
> > Weakness: If all the features are normalized, i.e., $|X_j|$, it seems to me that the feature that will correlate the most with $j$, will be $j$, and thus assumption 3 requires the missing entry to be observed. Could you comment on this?
>
> We are very sorry for the confusion created by a typo. Recall that $[p] = \\{1,...,p\\}$. Throughout the paper $\arg\max_{j \in [p]} \hat{\zeta}\_{i,j}$ and $\arg\max_{j \in [p]} \zeta\_{i,j}$ should be $\arg\max_{j \in [p] - \\{i\\}} \hat{\zeta}\_{i,j}$ and $\arg\max_{j \in [p] - \\{i\\}} \zeta\_{i,j}$. Note that therefore $i \neq j$.

---

### Review · Reviewer_GeJb · 2023-10-27

**Summary Of Contributions:**

This paper presents an efficient algorithm for missing value imputation using the topological property of the censorship filter. The authors provide theoretical guarantees for the exact recovery of the sparsity pattern using their imputation strategy, demonstrating that, under certain statistical and topological conditions, the underlying sparsity pattern can be consistently recovered with high probability.

**Audience:**

Yes

**Claims And Evidence:**

Yes

**Requested Changes:**

There are several areas of the paper that require significant improvement.

1. To enhance the experimental results, the authors should compare their method with a wider variety of imputation techniques, especially those relevant to the problem context as outlined in the paper. This would provide a more in-depth understanding of how the proposed method compares to existing techniques.

2. The authors should highlight more clearly the unique insights or techniques they are introducing in this field of research. If there are novel aspects to how they're applying the primal-dual witness construction approach to this particular problem, these should be emphasized and discussed in more detail.

3. The paper would be greatly enriched by a more comprehensive discussion of its limitations.

**Strengths And Weaknesses:**

Strengths:

The highlight on the theoretical guarantees for exact sparsity recovery and the establishment of sample complexity guarantees provides a clear indication of the paper's main contributions.

Weaknesses:

1. Comparison with other imputation methods: One significant concern is about experimental comparisons. The paper compares the proposed method with very simple imputation techniques. The paper indeed needs to compare its proposed method with more advanced or state-of-the-art imputation methods to convincingly demonstrate its effectiveness and superiority.

2. Theoretical contributions: While the paper is more theoretically-oriented, it is not clear what new contributions it is making in terms of theoretical techniques. The proof of the theoretical results is mainly based on primal-dual witness construction techniques, which have been developed well in previous work [1].

[1] Wainwright, M.J.. Sharp thresholds for High-Dimensional and noisy sparsity recovery using $\ell_{1} $-Constrained Quadratic Programming (Lasso). IEEE transactions on information theory, 55(5), pp.2183-2202, 2009.

---

> ### Author Response · Authors · 2023-11-15
>
> > The highlight on the theoretical guarantees for exact sparsity recovery and the establishment of sample complexity guarantees provides a clear indication of the paper's main contributions.
>
> We thank the reviewer for the appreciation of our work.
>
> > Weakness 1. ... The paper indeed needs to compare its proposed method with more advanced or state-of-the-art imputation methods to convincingly demonstrate its effectiveness and superiority.
> >
> > Change 1. ... the authors should compare their method with a wider variety of imputation techniques, especially those relevant to the problem context as outlined in the paper...
>
> We highlight that TMLR guidelines for reviewers specifically state "it should not be used as a reason to reject work ... because it isn't achieving a new state-of-the-art on some benchmark." We kindly refer to https://jmlr.csail.mit.edu/tmlr/reviewer-guide.html
>
> There are no prior techniques with provable guarantees for sparse regression with missing data. We are the first to provide support recovery guarantees in this scenario.
>
> Still, as suggested by Reviewer SnfY, we have tried MissForest and MICEForest in the experiment in Figure 3(b). MissForest and MICEForest are better than matrix completion, but slightly worse than our relatively simpler imputation technique. On the other hand, MissForest and MICEForest are far more complex to study theoretically. Thus, our relatively simpler imputation technique allows for good experimental results, together with a strong theoretical guarantee of support recovery.
>
> > Weakness 2. ... it is not clear what new contributions it is making in terms of theoretical techniques. The proof of the theoretical results is mainly based on primal-dual witness construction techniques, ... in previous work [1].
> >
> > Change 2. The authors should highlight more clearly the unique insights or techniques they are introducing in this field of research. If there are novel aspects to how they're applying the primal-dual witness construction approach to this particular problem, these should be emphasized and discussed in more detail.
>
> We are glad to have the opportunity to clarify the novelty of our work.
>
> The primal-dual witness framework has been used in several other works, which clearly did not prevent the other works from being published. This includes to mention a few, the analysis of:
> - sparse linear regression [Wainwright'09, Sharp Thresholds for High-Dimensional and Noisy Sparsity Recovery Using L1-Constrained QP]
> - learning Ising models [Ravikumar'10, High-Dimensional Ising Model Selection using L1-Regularized Logistic Regression]
> - learning Gaussian graphical models [Ravikumar'11, High-dimensional Covariance Estimation by Minimizing L1-penalized Log-determinant Divergence]
> - nonparametric regression [Ravikumar'07, Sparse Additive Models]
> - diffusion networks [Daneshmand'14, Estimating Diffusion Network Structures: Recovery Conditions, Sample Complexity & Soft-thresholding Algorithm]
>
> To run our algorithm, we first run the imputation step, and then we run the sparse regression step on the previously imputed data. Given this, it seems these two steps are somewhat independent, but they are in fact tightly related in the sense that our support recovery guarantee relies on both steps.
>
> Recall that:
> - $H_{i,j}$'s are the sample covariances computed from the observed entries only
> - $\zeta_{i,j}$'s are the neighbor scores
> - $\Pi(i)$'s are the top neighbor features
> - $\tau_i$'s are the error ratios
>
> All of the terms above are novel with respect to the primal-dual witness literature. Thus, all of our lemmas/theorems are novel, except the derivations in Section 4.2 and 4.3 (primal-dual witness), which on purpose we did not present as lemma/theorem, to avoid confusion about novelty.
>
> > Change 3. The paper would be greatly enriched by a more comprehensive discussion of its limitations.
>
> In Section 6 (Concluding Remarks) we highlight some limitations, but phrased them as future work:
> - We currently use the top neighboring feature to impute the missing value in our algorithm. As a future direction, this can be extended to the second, third, ..., neighboring features in a weighted fashion.
> - Another possible option is to take $y$ into account in the imputation step.
> - Moreover, it would be interesting to see if similar strategies and analysis will follow, for other supervised learning tasks masked by some censorship filters. Most of these problems, though commonly encountered in real world applications, do not have theoretical guarantees about imputation quality.
>
> Having said that, we would be happy to provide a more comprehensive discussion, expanding on the above.

---

> > ### Comment · Reviewer_GeJb · 2023-12-11
> >
> > Thank you for your response detailing the theoretical advancements made via the primal-dual witness construction in your manuscript. While I acknowledge the introduction of novel terms and lemmas/theorems, a key point requires further clarification to cement the non-trivial nature of your contributions.
> >
> > Please provide a concise yet comprehensive account of how the unique complexities of your problem have necessitated non-trivial adaptations of the primal-dual witness construction. I encourage you to succinctly detail these modifications, highlighting how they diverge from the traditional use and addressing the problem's unique challenges.

---

> > > ### Author Response · Authors · 2023-12-28
> > >
> > > Please recall that:
> > > - $H_{i,j}$'s are the sample covariances computed from the observed entries only
> > > - $\zeta_{i,j}$'s are the neighbor scores
> > > - $\Pi(i)$'s are the top neighbor features
> > > - $\tau_i$'s are the error ratios
> > >
> > > Theorem 1 provides a high probability statement for the empirical $\zeta_{i,\Pi(i)}$ to be better than all other empirical $\zeta_{i,j}$'s for $j \neq \Pi(i)$. As we mention in Theorem 1's statement "Consequently, the imputation result from the empirical score is consistent with the imputation result from the population score with high probability." (Please see also text at the beginning of Section 4.1.)
> > >
> > > Theorem 2 and 3: Remark 2 relates to bounding $z_{S^c}$ which is equal to $z^{(a)} + z^{(b)}$ in our manuscript. Theorem 2 bounds $z^{(a)}$ and Theorem 3 bounds $z^{(b)}$. Thus, Remark 2 applies to both Theorem 2 and Theorem 3.
> > >
> > > "Remark 2. Our proof relies on the careful analysis of the sub-Gaussian condition of the imputed matrix. Techniques from prior literature do not work in our model, because our missing structure $M$ is deterministic and cannot be reduced to some uniformly random noise. For instance, one classic technique is to write $X_{S^c}$ as a predictor of $X_S$ using the conditional covariance matrix (Wainwright, 2009a). This does not work in our case, because $\hat{X}$ (imputed matrix) will not cancel with the complement projection on $X$ (original matrix)."
> > >
> > > The following lemmas are technical developments needed for the above:
> > > - Lemma 1 (used in Theorem 1) provides a high probability statement for a ratio involving the empirical $H_{i,i}$.
> > > - Appendix Lemma 5 (used in Theorem 2) provides a high probability statement between the empirical and population $\tau_i$'s.
> > > - Lemma 2 (used in Theorem 3), Appendix Lemma 6 and Lemma 7 (used in Theorem 3) provide high probability statements for properties of the empirical $H$.

---

### Decision · Action_Editor_1Fn9 · 2024-01-10

**Recommendation:** Accept with minor revision

**Comment:**

This paper considers the problem of sparse linear regression with missing data, where the feature matrix is assumed to be incomplete.
The main contribution of this paper is the theoretical analysis to guarantee the successful recovery of support by a simple two-stage approach (imputing feature matrix with missing data + standard Lasso).

All three reviewers acknowledge the theoretical contributions of this paper and recommend its acceptance. The primary concerns raised by the reviewers pertain to the paper's lack of discussion with existing works and its writing clarity. During the author-reviewer discussion period, the authors merely responded to the reviewers without submitting a revision. Subsequently, the authors submitted a revised version addressing these comments. After reviewing the revision, I concur with the reviewers' recommendation to accept the paper. I also have additional two very minor comments regarding the writing and related work:
1. The title of section 4.3 does not seem to perfectly reflect its content. The beginning of section 4.3 is not well connected to the previous sections; the authors may consider adding a sentence such as, "To verify the strict dual feasibility condition (6), "

2. The authors may also briefly discuss the following related work on low-rank matrix completion with deterministic patterns:

M.C. Tsakiris. Low-rank matrix completion theory via Plucker coordinates, IEEE Transactions on Pattern Analysis and Machine Intelligence (PAMI), 2023.

**Audience:**

This paper would be of interest to researchers working on sparse regression and matrix completion (with deterministic patterns).

**Claims And Evidence:**

This paper considers the problem of sparse linear regression with missing data, where the feature matrix is assumed to be incomplete. Specifically, the sparsity pattern in the feature matrix is assumed to be deterministic rather than random. The paper first develops algorithms to impute the missing entries using their most significant observed neighboring feature and then applies the Lasso to perform sparse linear regression with the imputed feature matrix. This paper provides theoretical guarantees for the successful recovery of support by this two-stage method. The claims are supported by rigorous proof. The performance of the algorithm is also verified by synthetic experiments.